# Impact of the sampling procedure on the specific surface area of snow measurements with the IceCube

Julia Martin[1,2] and Martin Schneebeli[2]

[1]Alfred Wegener Institute, Helmholtz Centre for Polar and Marine Research (AWI), Telegrafenberg, 14473 Potsdam, Germany
[2]WSL Institute for Snow and Avalanche Research SLF, Flüelastrasse 11, 7260 Davos Dorf, Switzerland

**Correspondence:** Julia Martin (jmartin@outlook.de)

**Abstract.** The specific surface area (SSA) of snow is directly measured by X-ray computed tomography or indirectly using the reflectance of near-infrared light. The IceCube is a well-established spectroscopic instrument using a near-infrared wavelength of $1310\,\mathrm{nm}$. We compared the SSA of six snow types measured with both instruments. With the IceCube, we measured significantly higher values with a relative percentage difference between 20 to $52\,\%$ for snow types with an SSA between 5 to $25\,\mathrm{m^2\,kg^{-1}}$. We found no significant difference for snow with an SSA between 30 to $80\,\mathrm{m^2\,kg^{-1}}$. The difference is statistically significant between snow types but not uniquely related to the SSA. We suspected that particles artificially created during the sample preparation were the source of the difference. We sampled, measured and counted these particles to conduct numerical simulations with the radiation transfer solver TARTES. The results support the hypothesis that these small artificial particles can significantly increase the reflectivity at $1310\,\mathrm{nm}$ and, by that, lead to an overestimation of the SSA.

## 1 Introduction

The specific surface area of snow (SSA) is the most relevant parameter beyond the snow density for the structural characterisation of snow (Morin et al., 2013). The SSA is the surface of snow grains per mass in units of $\mathrm{m^2\,kg^{-1}}$ or over ice density in $\mathrm{m^{-1}}$. The SSA changes due to metamorphism and is about $100\,\mathrm{m^2\,kg^{-1}}$ for new snow and less than $2\,\mathrm{m^2\,kg^{-1}}$ for melt-freeze layers. SSA is relevant for snow chemistry, radiation transfer in snow, and mechanical properties ((Domine et al., 2008). For determining SSA indirectly, the IceCube (IC) is an efficient, transportable optical device operating at $1310\,\mathrm{nm}$ wavelength, which derives the SSA from the reflectance of snow. It uses a specific wavelength to invert from the reflectance of snow to the optical grain size of the snow (Domine et al., 2007; Gallet et al., 2014). Computed microtomography (micro-CT) is a high-resolution imaging technique to measure SSA from geometry (Kerbrat et al., 2008).

Coincidental measurements of the SSA of snow with both the IC and the micro-CT show inconsistent results, as first observed with a snow survey in Greenland in 2015 (Fig.1 (a)) at 72.57715°N and 38.45101°E (Schneebeli et al., 2015). Our study aims to elucidate the reasons for the observed differences. We focus here on artificial particles occurring on the snow sample surface produced with the mechanical IC sampling procedure as a source of the discrepancy. We investigate these particles' amount and size distribution in different snow types. We use a numerical simulation to study the particles' potential influence on the SSA measurement with the IC. Our study is the first systematic comparison of IC and micro-CT SSA measurements.

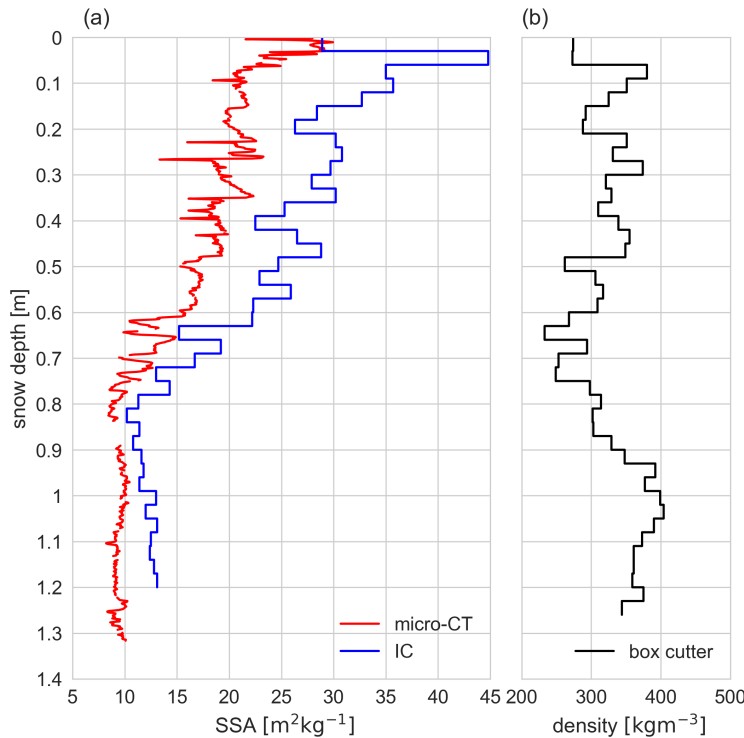

**Figure 1.** The data set with differing SSA measurement results for micro-CT and IC from the Greenland Summit expedition in 2015 at N72.577 15° and E−38.451 01° (Schneebeli et al., 2015). The snow surface is at $0\,\text{m}$. (a) is the entire SSA profile (about $1.3\,\text{m}$) measured with the micro-CT (red) and IC (blue). The IC has a $3\,\text{cm}$ measurement interval. (b) is the density profile (box cutter, volume $100\,\text{cm}^3$).

## 2  Methodology

### 2.1  Snow sampling and SSA measurements

We use five different alpine snow types for this study with an SSA between 5 to $25\,\text{m}^2\,\text{kg}^{-1}$. Type A (Fig.2 (a)) is a homogeneous alpine settled snow with a decomposed, rounded grain shape, size 0.5 to $1.5\,\text{mm}$, and an average density of $235 \pm 21\,\text{kg}\,\text{m}^{-3}$ (16 micro-CT samples). Type B (Fig.2 (b)) is the snow of type A, which we stored for 13 days at $-15\,°\text{C}$ in the cold lab under compacting weights. The grain shape is small rounded, and the grain size is medium with 0.5 to $1\,\text{mm}$, and the structure behaves more brittle than type A. Snow type B has an average density of $395 \pm 35\,\text{kg}\,\text{m}^{-3}$ (17 micro-CT samples). Type C (Fig.2 (c)) has large rounded, faceted shaped grains with a size between 1 and $2.5\,\text{mm}$ with an average density of $322 \pm 31\,\text{kg}\,\text{m}^{-3}$ (17 micro-CT samples. Type D (Fig.2 (d)) is snow with large rounded grains and size of 1 to $2\,\text{mm}$ and an average density of $432 \pm 43\,\text{kg}\,\text{m}^{-3}$ (18 micro-CT samples). Type E (Fig.2 (e)) is refrozen wet snow and has grains larger than $2.5\,\text{mm}$ and an average density of $302 \pm 67\,\text{kg}\,\text{m}^{-3}$.

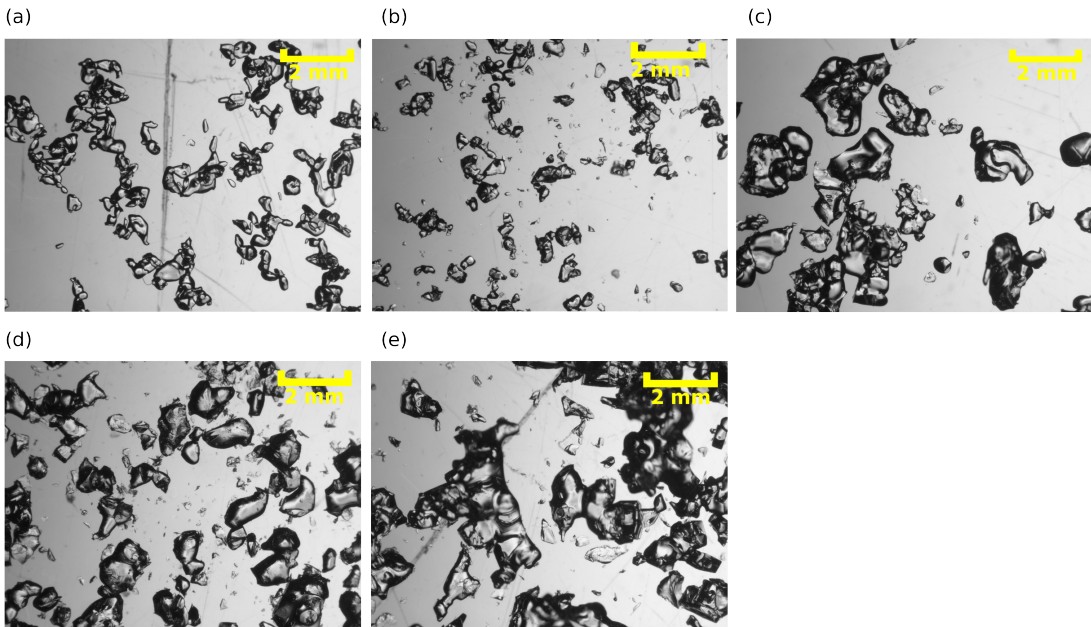

**Figure 2.** Pictures (Leica Z16 APO) of the undisturbed grain shape together with the artificial particles (fragments) for the five snow types we used for our sampling procedure. (a) rounded decomposed type A, (b) small rounded type B, (c) large rounded (facets) type C, (d) large rounded type D and (e) refrozen wet snow type E. The yellow scale bar is $2\,\mathrm{mm}$. We produced the pictures as part of our sampling procedure (see section 2.2).

Additionally, we consider a data set with an artificial snow type ($30 - 80\,\mathrm{m^2\,kg^{-1}}$) to cover a broad range of SSA. The artificial snow was produced following Brandt et al. (2011). This snow type is fine-grained snow and consists of spherical particles. It was produced in the cold laboratory at $-20\,^{\circ}\mathrm{C}$ room temperature by spraying cold water (Baumann, 2017). The samples were measured using the IC and micro-CT but not for the optical particle counting.

For our SSA sampling strategy for the IC and micro-CT, we used a homogeneous layer of approximately $6\,\mathrm{cm}$ within the snow block of each snow type (A, B, C, and D). To identify the homogeneous layer, we performed measurements with the SnowMicroPen version 2 (SMP) (Schneebeli and Johnson, 1998). Fig.3 shows the SMP profiles. Due to the highly heterogeneous character of snow type E, we did not obtain an SMP measurement and conducted our sampling procedure within the first $6\,\mathrm{cm}$ of the snow block. For snow types A, B, C and D, we cut off the unsuitable material above the homogeneous layer. We

brushed the surface gently to eliminate loose particles (Fig.4 (a)), which were produced by the sawing process.

     Our sampling procedure starts by retrieving an IC sample following the default method by Gallet et al. (2009) and described in the IC manual (A2 Photonic Sensors (2014)) (Fig.4 (b)). We transfer the snow sample from the piston, which is used to retrieve the snow from the snow block or snow cover, respectively, into the $60\,\mathrm{mm}$ diameter IC sampling holder (Fig.4 (c) and

(d)). We then tilt the sample and remove the protruding snow with a spatula (Fig.4 (e)) to create a flat surface for the SSA
measurement.

We measure the SSA with the IC in one orientation (Fig.7 IC + particles) and afterwards again tilt and gently tap the sample
over a Petri dish to remove any remaining loose particles from the surface (Fig.4 (f)). To study the loose particles, we weigh
the particle amount in the Petri dish (micro scales $\pm 0.01$ g) and photograph (Leica Z16 APO) the particles (see section 2.2 and
Fig.2). The mass of loose particles is divided by the surface area of the sample holder to receive the specific mass ($M_{spec}$).
Next, we perform a second IC SSA measurement (same sample) without loose particles on the sample surface (Fig.7 IC -
particles). For direct comparison, we take a standard micro-CT sample out of the middle of each IC sample (Fig.7 CT out
of IC ). We also retrieve standard micro-CT samples next to the IC samples within the undisturbed snow structure (Fig.7 CT
reference). To provide a meaningful study, we take between $5$ to $8$ IC samples per snow type and try to preserve this number
for each sampling step (Tab.1). Due to the fragility of, e.g. snow type C, the number of micro-CT samples is not constant.

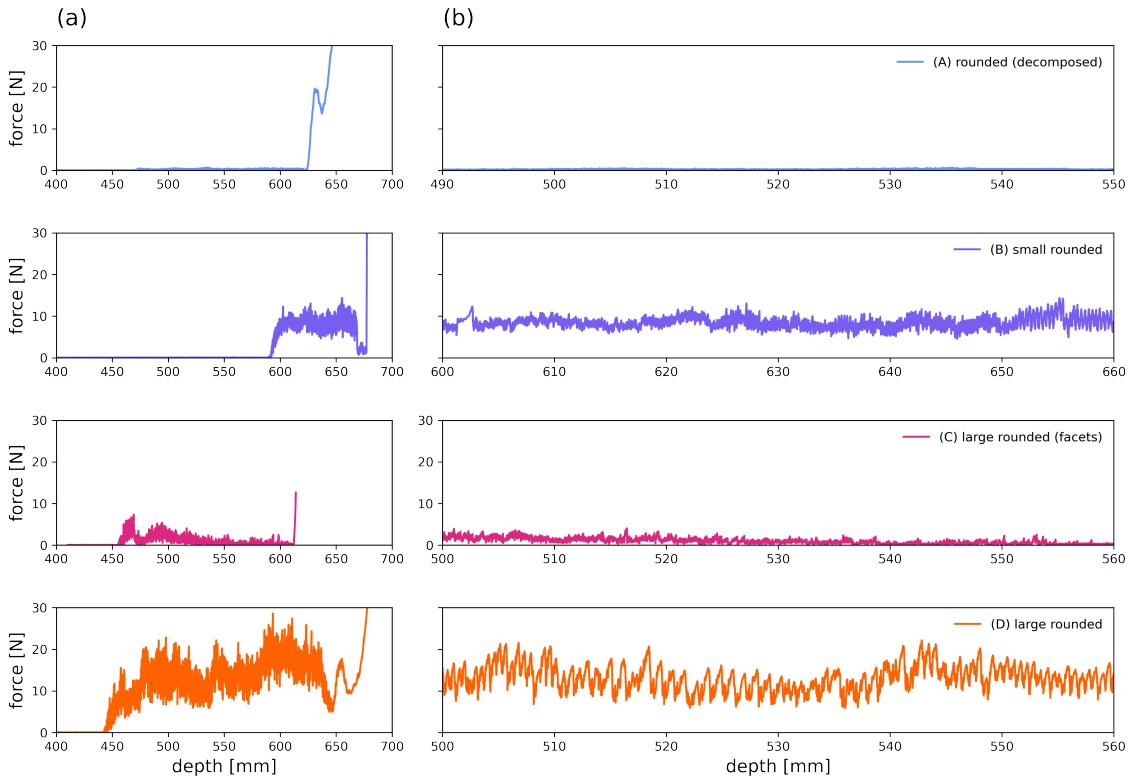

**Figure 3.** SMP measurements (Schneebeli and Johnson, 1998) for snow types A, B, C and D. Snow type E was unsuitable for SMP measurements due to its fragility and heterogeneity. (a) shows the force profile in [N] for the whole snow block. (b) displays the homogeneous layer (approximately $6$ cm within the snow block, which we use for our sampling procedure).

To reproduce the IC's mechanical sampling procedure, we use a specifically manufactured micro-CT sampling kit. The IC has a sampling kit with a 60 mm diameter holder. We manufactured a similar kit for the micro-CT that consists of a piston and a 30 mm diameter sample holder to improve the micro-CT resolution (Fig.5 (a)) and hence, is an imitation of the IC sampling kit. The same sampling steps are necessary to retrieve the sample from the snow block, including cutting the snow sample surface with a spatula. It allows scanning of in-situ broken particles on the surface produced during the IC-specific sampling

procedure (Fig.5 (b)), together with the snow structure deeper in the sample middle (Fig.5 (c)) with the micro-CT.

    We perform all micro-CT measurements with a computer tomography system working in a cold laboratory at $-15\,°C$ (Scanco Medical $\mu$CT40 or $\mu$CT80). For comparison with the IC, we measure samples with $\mu$CT40 (55 kVp energy, 15 $\mu$m nominal resolution, 145 $\mu$A intensity: type A; 55 kVp, 18 $\mu$m, 145 $\mu$A: B-E). The $\mu$CT40 has a carousel which allows us to measure up to 10 samples in a row. To measure the micro-CT sampling kit data set, we use the $\mu$CT80 (55 kVp, 15 $\mu$m, 145 $\mu$A:

A-E) which, only allows to measure one sample. We computed the SSA using the standard segmentation technique described in Hagenmuller et al. (2016).

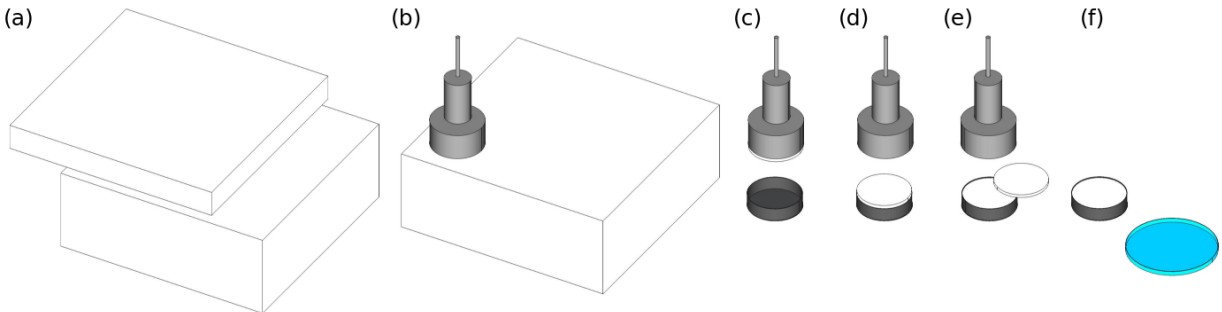

**Figure 4.** Step-by-step illustration of the sampling procedure. (a) within each snow block (snow types A, B, C and D), we identified a homogeneous layer of snow with SMP measurements. We removed the unsuitable material above the sampling layer with a saw. Afterwards, we gently brushed the surface to remove loose particles from the sawing process. (b) IC sample extraction with the IC sampling device (piston). The snow sample is 35 mm thick. (c) transfer of the snow sample into the IC sampling holder, which has a height of 30 mm. (d) transferred snow sample with 5 mm of protruding snow. (e) we cut off the protruding snow with a sharp spatula (the sample is slightly tilted) following (Gallet et al., 2009) and perform the first IC measurement (IC + particles). (f) we tilt the IC sampling holder above a Petri dish (blue) to collect the remaining loose particles created during the sampling step (e) for further analysis (macro-photographs and particle weight) and the sample is measured again with the IC (IC - particles).

## 2.2   Grain survey

    We examine and photograph the loose particles created by IC sample preparation in a Petri dish to study their influence on the surface of the IC sample. The Petri dish is subdivided into eight sections and eight circles. By generating random numbers,

we receive eight coordinates per sample, resulting in 40 pictures per snow type sample. This procedure guarantees unbiased

pictures without a particular preference for the grain shape. To obtain the grain size distribution, we randomly choose eight pictures with about 1000 grains in total for snow types B, D, and E. For snow types A and C, we choose 12 pictures to obtain the same number of grains.

We measure the length and width of every particle with ImageJ (Schneider et al., 2012) and calculate the optical diameter $d_{opt}$ and SSA ($SSA_{mass}$), assuming the main grain shape to be ellipsoidal. We record the total amount of measured grains, the median diameter, the median SSA, the optical diameter $d_{opt}$ and the $M_{spec}$ for each snow type in Tab.1.

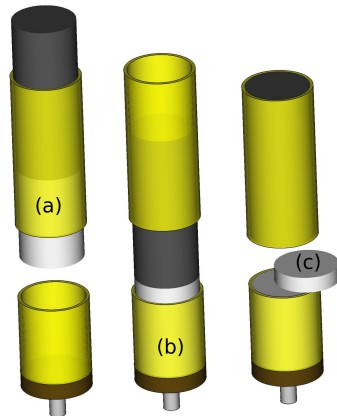

**Figure 5.** Illustration of the micro-CT sampling kit with sample holder in yellow and piston in black. The snow sample is coloured in white. (a) is the manufactured micro-CT sampling kit with the piston imitation (35 mm depth) to retrieve the snow sample. (b) illustrates the snow sample transfer into the sampling holder (30 mm depth), and (c) shows the cutting process of protruding snow to create a flat sample surface.

## 2.3 TARTES simulations

To simulate the influence of a layer of artificial particles on top of a substrate snow layer, we use the TARTES model (Two-streAM Radiative TransfEr in Snow) (Libois et al., 2013). The SSA of the particle layer on top is the median SSA we calculated from the grain examination for each snow type. The density of the particle layers ($\rho_{layer}$) is set to $200\,\mathrm{kg\,m^{-3}}$.

We use the average SSA ($SSA_{substrate}$, Tab.1) and average density ($\rho_{substrate}$, Tab.1) of the micro-CT samples out of the IC samples (CT out of IC) as input for the substrate layer below and the synthetic substrate depth is set to $1\,\mathrm{m}$. In the simulations, we use $1310\,\mathrm{nm}$ wavelength to study the particle influence of the IC operating wavelength and $950\,\mathrm{nm}$ wavelength as an approximation to the NIR photography. For the first step, we calculate the diffuse albedo for the two-layer simulation using the albedo function of the TARTES python module. To calculate the SSA, we needed to apply a conversion to the diffuse albedo data set. Hence, we produced a synthetic SSA data set for both wavelengths, which we use for a poly-fitting procedure. We fit a five-degree polynomial adequate to describe the relationship between the calculated diffuse albedo and SSA.

## 3 Results

Fig.6 (a) shows that the SSA of the artificial snow is the same for both the IC and CT. We measure a lower SSA with the
micro-CT for all sintered snow types, independent of the SSA, compared to the IC (Fig.6 (b), details in Tab.1 and Fig.7). We
also measure a lower SSA with the micro-CT if we scan the prepared flat surface compared to the undisturbed middle of the
sample (Fig.6 (c)) using the micro-CT sampling kit. We now present the detailed results for the different treatments of the
samples.

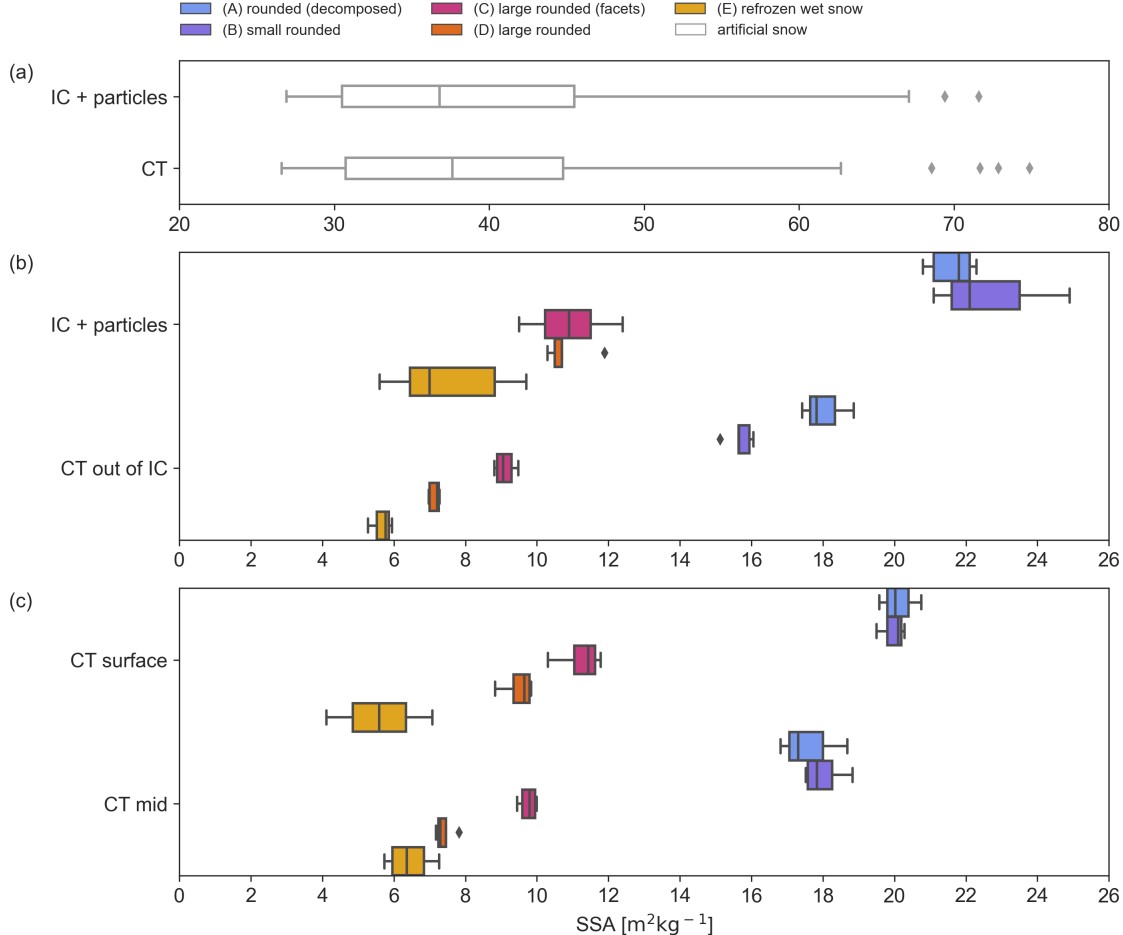

**Figure 6.** All SSA measurements are classified in sampling steps. The boxplots represent the median and interquartile range. IC + particles
is the default method (Gallet et al., 2009; A2 Photonic Sensors, 2014). CT is the micro-CT measurements out of an IC + particles sample. CT
out of IC is the micro-CT measurements out of an IC - particles sample. CT surface and CT mid refer to the manufactured micro-CT sampling
kit. (a) are the results for the artificial snow data set with SSA in the range of 30 to $80\,\mathrm{m^2\,kg^{-1}}$. (b) shows the 5 snow types (A,B,C,D,E).
The SSA is between 5 to $25\,\mathrm{m^2\,kg^{-1}}$. (c) are the results for the micro-CT sampling kit with SSA in the range of 5 to $22\,\mathrm{m^2\,kg^{-1}}$. All results
can be found in Tab.A1. For the data set, see code and data availability.

The artificial snow has no significant difference between the methods (n = 45, p-value: .43, p<0.05). The sintered snow types A, B, C and D show a significant difference at the $0.05$-level between IC and micro-CT ($\overline{SSA}$ (IC + particles) vs $\overline{SSA}$ (CT out of IC), see Tab.1 and Fig.7). Snow type E (four micro-CT samples) had a larger scatter both in IC and micro-CT and is therefore not statistically different. However, the SSA is 24% smaller measured by the micro-CT compared to the IC.

The treatment of "removing loose particles" (IC - particles) leads to a slight decrease in the IC measurements. The difference between these two sampling steps (IC + particles vs IC - particles, Tab.1) is not significant for types A, B and E and is significant for types C and D. Comparing these (IC - particles) with micro-CT samples (CT out of IC), the difference is not significant for types A, B, C and D. The difference between the micro-CT samples (CT out of IC) and the micro-CT samples taken out of the same snow block next to the IC samples (CT reference) is not significant for types A, B, and E. There is only one reference sample for type C.

Most surface samples measured with the micro-CT sampling kit (Fig.6c and Fig.8) show a higher SSA than those in the undisturbed middle of each sample. The difference is significant for types C and D. For types A and B, the SSA measured at the samples' surface is higher than those measured in the middle, but not significantly. The SSA at the surface is lower than in the middle for the heterogeneous refrozen wet snow type E. Type D shows the highest mean SSA difference with 28%. Fig.8 shows the three-dimensional reconstruction of this particular snow sample. Fig 8 (a) is the sample surface, and Fig.8 (b) displays the middle. The fragmented artificial particles (less than $0.5\,\mathrm{mm}$) protrude compared to the undisturbed snow structure with a size of $1\,\mathrm{mm}$ and more.

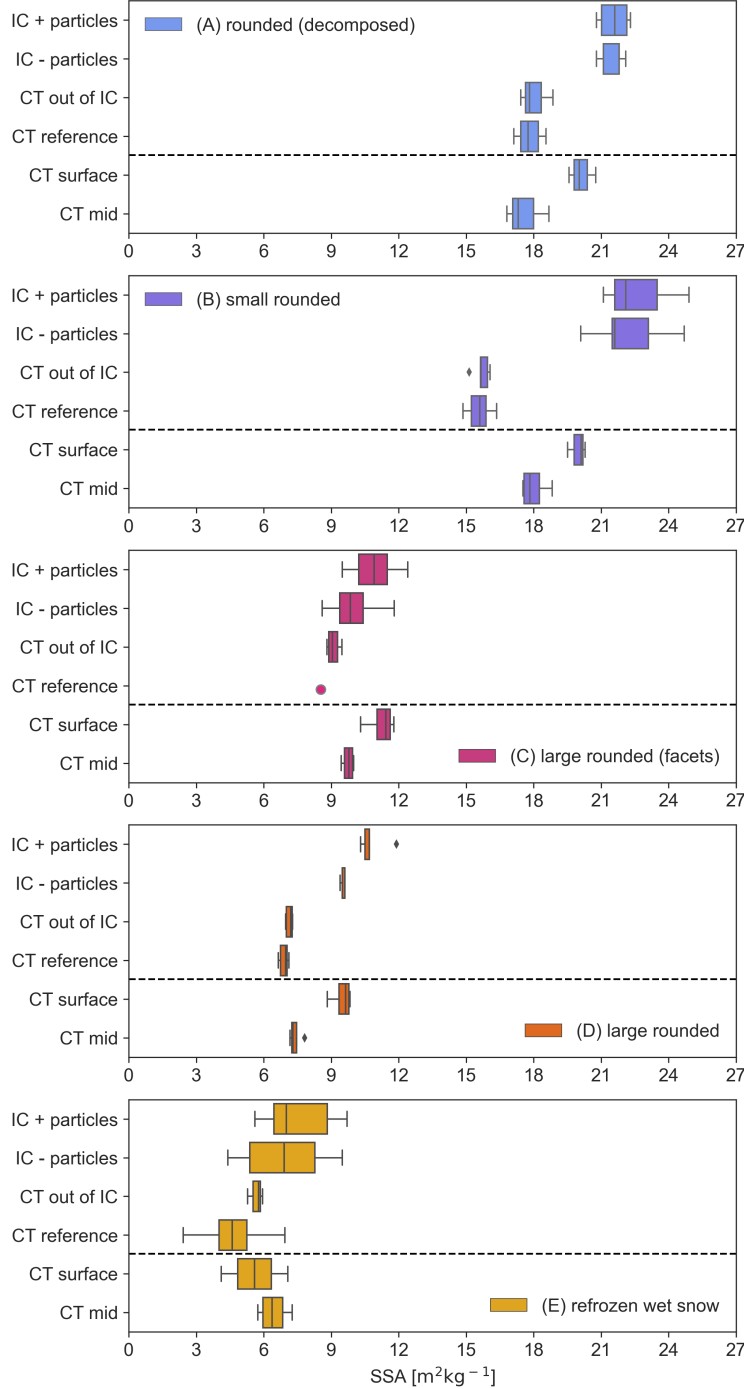

**Figure 7.** All SSA measurements classified in snow types and subdivided into sampling steps. SSA in range of 5 to $25\,\mathrm{m^2\,kg^{-1}}$. IC + particles is the default sampling process (Gallet et al., 2009). Results for the micro-CT sampling kit separated by the dotted grey line in each plot. SSA in range of 5 to $22\,\mathrm{m^2\,kg^{-1}}$. The results can be found in Tab.1.

Table 1. Means, standard deviations and number of measurements (n) for all measurements organized following the multi-levelled sampling strategy.

IC + particles represents the default sampling procedure suggested by Gallet et al. (2009).

The line with $\rho^*_{layer}$ is the synthetic density used for the TARTES simulations. For data set: code and data availability.

| | Unit | A | (n) | B | (n) | C | (n) | D | (n) | E | (n) |
|---|---|---|---|---|---|---|---|---|---|---|---|
| $\overline{SSA}$ (IC + particles) | $m^2\,kg^{-1}$ | $21.6 \pm 0.6$ | (5) | $22.6 \pm 1.5$ | (5) | $10.9 \pm 1$ | (8) | $10.8 \pm 0.6$ | (5) | $7.5 \pm 1.7$ | (6) |
| $d_{opt}$ (IC + particles) | $\mu m$ | 303 | | 289 | | 603 | | 605 | | 872 | |
| $\overline{SSA}$ (IC - particles) | $m^2\,kg^{-1}$ | $21.5 \pm 0.5$ | (5) | $22.2 \pm 2.8$ | (5) | $10 \pm 1$ | (8) | $9.5 \pm 0.1$ | (5) | $6.9 \pm 2$ | (6) |
| $d_{opt}$ (IC - particles) | $\mu m$ | 304 | | 295 | | 661 | | 687 | | 951 | |
| $\overline{SSA}$ (CT out of IC) | $m^2\,kg^{-1}$ | $18 \pm 0.6$ | (5) | $15.7 \pm 0.4$ | (5) | $9.1 \pm 0.3$ | (8) | $7.1 \pm 0.1$ | (5) | $5.7 \pm 0.3$ | (3) |
| $d_{opt}$ (CT out of IC) | $\mu m$ | 363 | | 417 | | 720 | | 916 | | 1155 | |
| $\overline{SSA}$ (CT reference) | $m^2\,kg^{-1}$ | $17.8 \pm 0.6$ | (5) | $15.6 \pm 0.6$ | (5) | $8.5$ | (1) | $6.9 \pm 0.2$ | (5) | $4.6 \pm 1.7$ | (5) |
| $d_{opt}$ (CT reference) | $\mu m$ | 367 | | 420 | | 767 | | 948 | | 1409 | |
| $\rho_{mean}$ (CT) | $kg\,m^{-3}$ | $235 \pm 22$ | (16) | $395 \pm 35$ | (17) | $322 \pm 31$ | (17) | $432 \pm 43$ | (18) | $302 \pm 67$ | (14) |
| **Micro-CT sampling kit** | | | | | | | | | | | |
| $\overline{SSA}$ (CT surface) | $m^2\,kg^{-1}$ | $20.1 \pm 0.6$ | (3) | $20 \pm 0.4$ | (3) | $11.2 \pm 0.7$ | (4) | $9.5 \pm 0.5$ | (4) | $5.6 \pm 2.1$ | (2) |
| $d_{opt}$ (CT surface) | $\mu m$ | 325 | | 328 | | 582 | | 689 | | 1170 | |
| $\overline{SSA}$ (CT mid) | $m^2\,kg^{-1}$ | $17.6 \pm 1$ | (3) | $18 \pm 0.5$ | (4) | $9.8 \pm 0.3$ | (4) | $7.4 \pm 0.3$ | (4) | $6.4 \pm 0.7$ | (4) |
| $d_{opt}$ (CT mid) | $\mu m$ | 372 | | 363 | | 671 | | 884 | | 1017 | |
| **Grain survey** | | | | | | | | | | | |
| grains (measured) | | 500 | | 1582 | | 997 | | 1962 | | 1130 | |
| $\overline{M}_{part}$ | g | $0.11 \pm 0.02$ | (5) | $0.22 \pm 0.05$ | (5) | $0.72 \pm 0.26$ | (8) | $0.85 \pm 0.17$ | (5) | $2.22 \pm 0.97$ | (6) |
| $SSA_{med}$ (grain survey) | $m^2\,kg^{-1}$ | $14 \pm 5$ | | $49 \pm 32$ | | $72 \pm 57$ | | $85 \pm 58$ | | $61 \pm 36$ | |
| $d_{opt}$ (grain survey) | $\mu m$ | $461 \pm 172$ | | $133 \pm 92$ | | $91 \pm 64$ | | $77 \pm 49$ | | $107 \pm 65$ | |
| specific mass | $kg\,m^{-2}$ | 0.04 | | 0.08 | | 0.25 | | 0.3 | | 0.8 | |
| **TARTES** | | | | | | | | | | | |
| $SSA_{layer}$ | $m^2\,kg^{-1}$ | 14 | | 49 | | 72 | | 85 | | 61 | |
| $\rho^*_{layer}$ | $kg\,m^{-3}$ | 200 | | 200 | | 200 | | 200 | | 200 | |
| $SSA_{substrate}$ | $m^2\,kg^{-1}$ | 18 | | 16 | | 9 | | 7 | | 6 | |
| $\rho_{substrate}$ | $kg\,m^{-3}$ | 233 | | 420 | | 326 | | 449 | | 312 | |

## 3.1 Grain counting

We examined the size and number of the particles (Tab.1) formed at the surface of the micro-CT sampling kit. Tab.1 contains the counted number of grains and the weight of particles in the Petri dish. The weight of particles, all measured on the same surface area, varies between $0.11\,\mathrm{g}$ to a maximum of $2.22\,\mathrm{g}$, corresponding to a specific mass of $0.04\,\mathrm{kg\,m^{-2}}$ and $0.8\,\mathrm{kg\,m^{-2}}$, respectively. The calculated particle median SSA ($\mathrm{SSA}_{med}$ (grain survey), Tab.1) is the lowest for type A ($14 \pm 5\,\mathrm{m^2\,kg^{-1}}$) and the highest for type D ($85 \pm 58\,\mathrm{m^2\,kg^{-1}}$). It is $49 \pm 32\,\mathrm{m^2\,kg^{-1}}$ for type B, $72 \pm 57\,\mathrm{m^2\,kg^{-1}}$ for type C and $61 \pm 36\,\mathrm{m^2\,kg^{-1}}$ for type E. All snow types have a left-skewed particle size distribution. The median optical diameter ($\mathrm{d}_{opt}$ (grain survey)) is between $77 \pm 49\,\mathrm{\mu m}$ to $133 \pm 92\,\mathrm{\mu m}$, except for type A ($461 \pm 172\,\mathrm{\mu m}$).

We found an increasing ratio between the optical diameter of the CT reference and the particles as follows: Type A: 0.80, B: 3.2, C: 8.4, D: 12.3, E: 13.2. The optical diameter of the particles is smaller than the optical diameter of the undisturbed snow, except for the rounded snow (Type A). The ratio and the size show that the sample preparation leads to new small ice particles at the surface. The larger particles of rounded snow (Type A) are just at the limit of statistical significance.

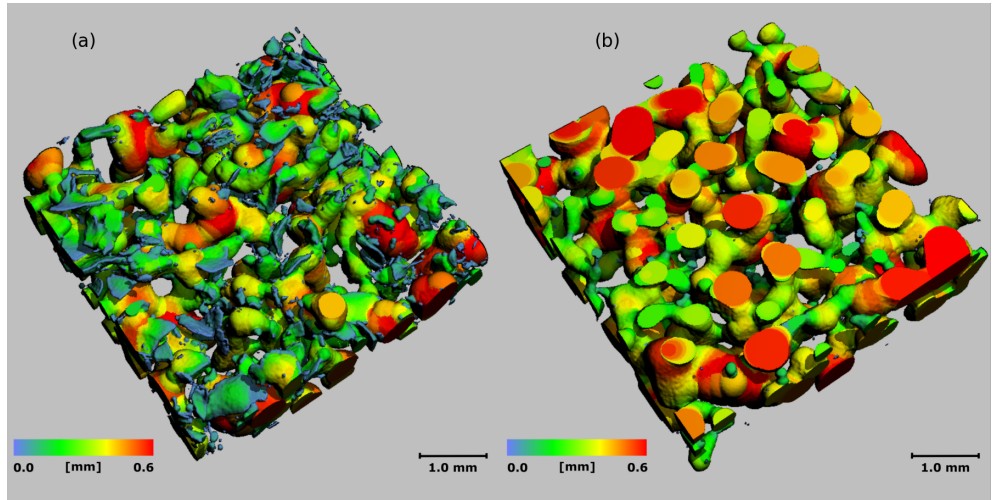

**Figure 8.** 3D reconstruction for a sample of snow type D taken with the micro-CT sampling kit (Fig.5). We performed those measurements with the $\mu$CT80 ($55\,\mathrm{kVp}$, $15\,\mathrm{\mu m}$, $145\,\mathrm{\mu A}$). (a) is the sample surface scan with artificial particles ($<0.5\,\mathrm{mm}$) prepared following the default IC sampling procedure. (b) are the snow grains ($>0.5\,\mathrm{mm}$) scanned in the middle of the sample without the influence of mechanical sampling treatment.

## 3.2 TARTES

Fig. 9 (a) displays the influence of the converted SSA with increasing particle layer thickness. Fig.9 (b) shows the percentage change in SSA with increasing particle thickness. Both layers have different SSA and density values ($SSA_{substrate}$ and $\rho_{substrate}$, Tab.1).

Our TARTES simulations (Fig.9a) show the SSA at $1310\,\mathrm{nm}$ and $950\,\mathrm{nm}$ wavelengths. Figure 9b shows the percentage change in SSA for $1310\,\mathrm{nm}$ and $950\,\mathrm{nm}$ over increasing particle layer thickness. The SSA for $1310\,\mathrm{nm}$ shows a steeper increase with layer thickness compared to $950\,\mathrm{nm}$. The SSA of snow type A shows a slight negative trend with an increase in thickness because the particle layer SSA is smaller than the substrate SSA. The plot on the right shows the percentage change in SSA for $1310\,\mathrm{nm}$ and $950\,\mathrm{nm}$ over increasing particle layer thickness. The starting point at 100% marks the pure substrate SSA for each snow type. The SSA measured at a wavelength of $1310\,\mathrm{nm}$ is much more susceptible to the influence of a layer with loose particles. The changes are most prominent for snow types D and E. The slope is about four times higher at $1310\,\mathrm{nm}$ than at $950\,\mathrm{nm}$ wavelength.

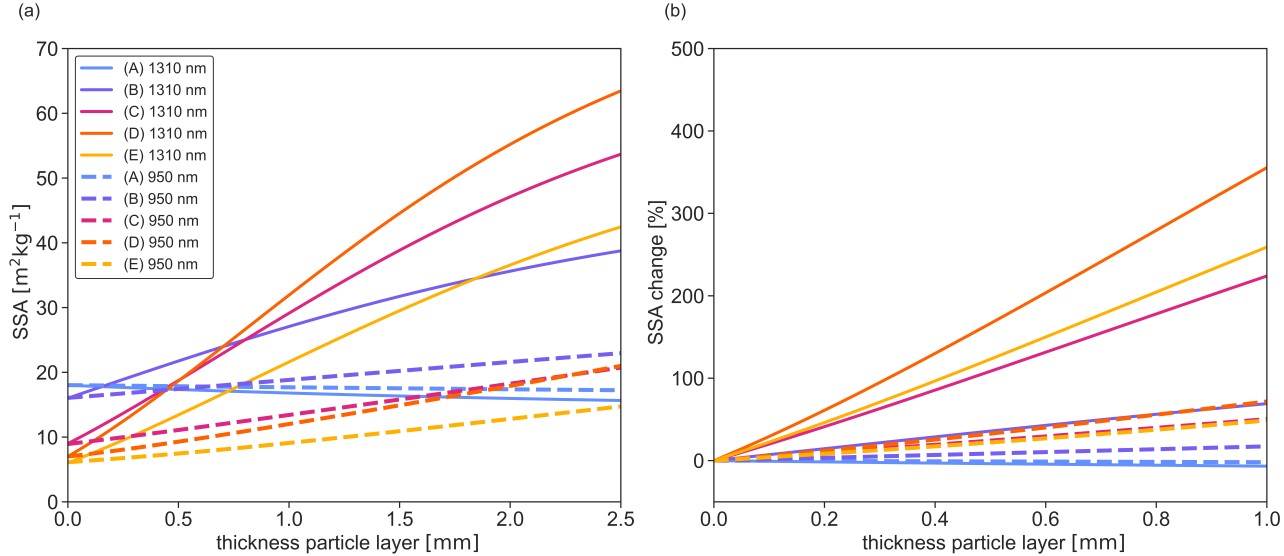

**Figure 9.** Simulation of a $1\,\mathrm{m}$ thick snow substrate with a thin ($< 2.5\,\mathrm{mm}$) particle layer on top with a density of $200\,\mathrm{kg\,m^{-3}}$. Both layers have different SSA values (Tab.1). The TARTES simulations (a) show the SSA at $1310\,\mathrm{nm}$ and $950\,\mathrm{nm}$ wavelength. (b) shows the percentage change in SSA for 1310 and $950\,\mathrm{nm}$ over increasing particle layer thickness.

## 4  Discussion

We found systematic differences between the IC and the micro-CT SSA measurements for snow with an SSA between $5\,\mathrm{m^2\,kg^{-1}}$ to $25\,\mathrm{m^2\,kg^{-1}}$. The most significant difference appears when the sample is prepared following the default sampling procedure. The relative percentage difference ranges from $52\,\%$ for type D with a Mean Bias Error (MBE) of $3.7\,\mathrm{m^2\,kg^{-1}}$ ($44\,\%$ and MBE $7\,\mathrm{m^2\,kg^{-1}}$ type B, $32\,\%$ and MBE $2.2\,\mathrm{m^2\,kg^{-1}}$ type E) to $20\,\%$ for type A (MBE $3.8\,\mathrm{m^2\,kg^{-1}}$) and C (MBE $1.8\,\mathrm{m^2\,kg^{-1}}$). It can be decreased down to $34\,\%$ for type D ($41\,\%$ type B, $21\,\%$ type E, $19\,\%$ type A) to $9\,\%$ for type C by removing loose particles. In contrast, we did not find a significant difference in the artificial snow data set with an SSA between $39\,\mathrm{m^2\,kg^{-1}}$ to $80\,\mathrm{m^2\,kg^{-1}}$.

We conclude that the observed difference depends on the predisposition of the snow type to produce small surface particles and is not linked to the SSA. We show that small ice particles form during the flat surface preparation for IC measurements. These particles are preferentially formed from the bonds, as there seems to be a close link to sintering strength. The small particles of the disaggregated snow change the optical properties of the surface and consequently lead to an overestimation of the SSA. It partly corresponds with the observations done by Gallet et al. (2009). They found an overestimation of up to $5\,\%$ for hard wind slab layers and suggested brushing the surface to remove particles falsifying the measurement. Gallet et al. (2009) detected an overall accuracy in SSA measurements better than $12\,\%$ for the DUFISS device (pre-production model of the IC). Our results indicate that measurements at the wavelength of 1310 nm are prone to systematic biases for most snow types.

The micro-CT sampling kit works well to reconstruct the IC sampling procedure, and the potential layer of artificial particles on the sample surface became visible (Fig.8). The discrepancies between the surface and the sample centre in the range of $11\,\%$ (type B) to $28\,\%$ (type D) support the inference that the number of small particles produced is different for each snow type. The size distributions show most particles to be smaller than $133\,\mu\mathrm{m}$ for types B, C, D and E, which does not correspond with the grain size of the original snow. For each snow type except type A, the survey reveals an optical diameter at least half the size (type B: $133\,\mu\mathrm{m}$ ) compared to the optical diameter estimated from the SSA measured with the IC (type B: $289\,\mu\mathrm{m}$) and at least three time times smaller than the optical diameter from the CT measurements (type B: $417\,\mu\mathrm{m}$). For snow of types C, D and E, the $d_{opt}$ is $54\,\%$ (type B) up to $88\,\%$ (type E) smaller compared to measurements with the IC and $68\,\%$ (type B) up to $91\,\%$ smaller compared to the CT out of IC measurements. The rounded grain of snow type A appears to be very fragile and does not produce small particles. Consequently, the broken particles are very similar in size to the undisturbed snow.

The TARTES simulations show that the albedo at the $950\,\mu\mathrm{m}$ wavelength is more robust and less susceptible towards an artificial particle layer than the albedo at $1310\,\mu\mathrm{m}$. The simulations with $1310\,\mu\mathrm{m}$ explain the deviation in SSA between the IC and the micro-CT reasonably. For the more fragile snow types C, D and E, a particle layer thickness of less than $0.5\,\mathrm{mm}$ bridges the deviation. A $1\,\mathrm{mm}$ thick particle layer can explain the deviation for the less sensitive snow type B. Type A shows no effect of a particle layer. The reason is that the e-folding depth at 950 nm is about twice than at 1310 nm. Gallet et al. (2009) choose 1310 nm as the absolute reflectance difference for the relevant SSA is larger at this wavelength than at 950 nm. However, the precision of the measurement depends not only on the absolute reflectance difference, but equally to the signal-noise ratio of the receiver, either a photodiode (in the case of a point instrument) or the CCD (in the case of a camera).

We finally conclude that the measured differences in SSA of the strongly sintered snow from the Greenland Summit expedition (Fig.1) are explained by the formation of small particles during IC-sample preparation.

## 5  Conclusions

The key to explaining the deviation between IC and micro-CT SSA measurements is the mechanical destruction of parts of the original snow structure into small, artificial particles during the sample preparation process. Those particles lead to an SSA overestimation by the IC. The formation of these particles is related to different variables, such as SSA, temperature, grain shape and sampling treatment. Fresh, poorly bonded snow with a high SSA shows a negligible effect, to a lesser degree,

and also rounded snow. The measured difference between micro-CT and IC is however not limited to a specific geographic origin of the snow. We can simulate the effect of the SSA bias with TARTES simulations. The formation of small particles at the surface is a systematic source of bias in spectroscopic measurements. We found that the IC overestimated the SSA and underestimated the optical diameter for most snow types in our study. Additional sample surface preparation steps, such

brushing or vacuuming, could be applied, but the outcome and benefits are uncertain and need further investigation.

*Code and data availability.* The data set and the source code for the TARTES simulation are available on doi:10.16904/envidat.333.

# Appendix A

**Table A1.** Summary of the statistical analyses. The first step was to test for normal distribution. Then testing for significant differences between the sampling steps. The subscript $a$ marks data sets for which at least one is not normally distributed but is assumed for t-testing. IC + particles represents the default sampling procedure suggested by Gallet et al. (2009).

| Snow type | A | B | C | D | E |
|---|---|---|---|---|---|
| Test normal distribution | | | | | |
| IC + particles | Yes | Yes | Yes | Yes | Yes |
| IC - particles | Yes | Yes | Yes | Yes | Yes |
| CT out of IC | Yes | Yes | Yes | Yes | No |
| CT reference | Yes | Yes | No (n<2) | Yes | Yes |
| CT surface | No (n<5) | No (n<5) | Yes | Yes | No (n<3) |
| CT mid | No (n<5) | Yes | Yes | Yes | Yes |
| T-test (one-tailed) | | | | | |
| IC + particles IC - particles | p-value is .40 not significant at p<0.05 | p-value is .34 not significant at p<0.05 | p-value is .05 significant at p<0.05 | p-value is .00 significant at p<0.05 | p-value is .29 not significant at p<0.05 |
| IC + particles CT out of IC | p-value is .00 significant at p<0.05 | p-value is .00 significant at p<0.05 | p-value is .00 significant at p<0.05 | p-value is .00 significant at p<0.05 | p-value is .05 not significant at p<0.05$_a$ |
| IC - particles CT out of IC | p-value is .00 significant at p<0.05 | p-value is .00 significant at p<0.05 | p-value is .01 significant at p<0.05 | p-value is .00 significant at p<0.05 | p-value is .17 not significant at p<0.05$_a$ |
| CT out of IC CT reference | p-value is .30. not significant at p<0.05 | p-value is .38 not significant at p<0.05 | none | p-value is .03 significant at p<0.05 | p-value is .17 not significant at p<0.05$_a$ |
| CT surface CT mid | p-value is .01 significant at p<0.05$_a$ | p-value is .00 significant at p<0.05$_a$ | p-value is .00 significant at p<0.05 | p-value is .00 significant at p<0.05 | p-value is .23 not significant at p<0.05$_a$ |

| T-test) (one-tailed) | artificial snow data set | IC CT | p-value is .43 not significant at p<0.05 | | |
|---|---|---|---|---|---|

**Table A2.** Table with equations used in this paper, including parameters and units.

| | Parameter | Unit | Equation | Information |
|---|---|---|---|---|
| Weight of particles in Petri dish | $M_{part}$ | kg | | measured with scale |
| Optical diameter for particles (assumed to be ellipsoidal) | $d_{opt}$ | m | $\frac{(a+c)}{2}$ | $a$: sphere length $c$: sphere width |
| Surface of particles | $A_{part}$ | m$^2$ | $4\pi r_{opt}^2$ | |
| Volume of particles | $V_{part}$ | m$^3$ | $\frac{4}{3}\pi r_{opt}^3$ | |
| Specific mass particle layer | $M_{spec}$ | kg m$^{-2}$ | $\frac{M_{part}}{A_{holder}}$ | $r_{holder} = 0.03\,\text{m}$ |
| SSA of particles | $SSA_{mass}$ | m$^2$ kg$^{-1}$ | $\frac{A_{part}}{\rho_{ice}V_{part}}$ | |
| Optical diameter | $d_{opt}$ | m | $\frac{6}{\rho_{ice}SSA_{mass}}$ | |

*Author contributions.* JM and MS designed the study and conceptualized the data collection. JM carried out the measurements and the simulations and wrote the paper. JM and MS interpreted the results. JM wrote the first draft, and MS contributed to the paper.

*Competing interests.* The authors declare that they have no conflict of interest.

*Acknowledgements.* Our sincere gratitude is with Henning Löwe for his ongoing support, enthusiasm and valuable comments. We would like to thank Martin Proksch, Margret Matzl and Lino Schmid for the Greenland Summit 2015 data set contribution. Furthermore, we appreciate the work of Philipp Baumann who provided the artificial snow data set.

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
