# Peer review of "Impact of the sampling procedure on the specific surface area of snow measurements with the IceCube"

_EGUsphere, 2022_

## Author Response (AR1)

**Author Response**

Paper # https://doi.org/10.5194/egusphere-2022-501

**Impact of the sampling procedure on the specific surface area of snow measurements with the IceCube**

Martin and Schneebeli

**Response Review RC1**

This manuscript is potentially valuable as it contains an important headline result – that the manner in which snow is sampled impacts up to approximately 50% the SSA values less than 30 m2 kg-1 from the IceCube (IC) instrument in comparison to micro-CT (CT). Potentially, this has big implications for 1310 nm snow reflectance measurements as the IceCube (n.b. also potentially DUFISSS in pre-production, or IRIS in non-commercial form), is an increasingly common and robust in-field instrument for objective measurements of snow microstructure. Micro-CT, a 'gold standard', is as good a direct measure of snow microstructure as we currently have. Behind this headline result, there are a number of issues that need to be addressed for the community to have confidence in the currently proposed message. It may mean the scope of the message needs to be refined with greater detail, and the implications limited to particular snow types.

The 5-25 m2 kg-1 range of snow SSA that shows significant different between IC and CT are often associated with important snow types, e.g. depth hoar or wind slab in Arctic and sub-Arctic snowpacks, that are not part of the experiment. At best this paper needs to be explicitly limited to Alpine snow, otherwise unintended mis-interpretation could occur. Could more detail be provided to describe the Alpine snow types measured, e.g. densities from volumetric sampling? More details of the snowpack from which the samples were extracted would be highly beneficial so the reader can get a feel for snow types which the interpretation is both relevant for and limited to.

We included SnowMicroPen (SMP) measurements for snow types A, B, C and D. The force [N] profiles show the homogeneity of the sampled layer within the snow block (Fig. 3). Additionally, we added the densities for snow types A, B, C, D and E measured with the micro-CT.

Manuscript Fig.3. SMP measurements (Schneebeli and Johnson, 1998) for snow types A, B, C and D. Snow type E was unsuitable for SMP measurements due to its fragility and heterogeneity. (a) shows the force profile in [N] for the whole snow block. (b) displays only the homogeneous layer (approximately 6 cm within the snow block, which we use for our sampling procedure).

Furthermore, we included pictures of the grain shape and size for snow types A, B, C, D and E (see Fig. 2) to provide more information about the snow types we used for our sampling strategy.

Manuscript Fig.2. Pictures of the grain size and shape for the five snow types we used for our sampling procedure. (a) rounded decomposed A, (b) small rounded B, (c) large rounded (facets) C, (d) large rounded D and E refrozen wet snow E. The yellow scale bar is 2 mm.

The sample preparation process is a key conclusion explaining the difference between IC and CT. However, there is ambiguity in the description of this method. It seems the sample was reduced to size through cutting of unsuitable material and then brushed gently to remove loose particles and measured by IC. Secondly, the 'default method following Gallet et al. (2009)' was followed, then any remaining loose particles were knocked off, then the sample was remeasured using IC. The default method in section 2 of Gallet et al. (2009) refers to the sample measurement face being shaved off with a spatula, in which they state it was difficult to obtain a 'perfect surface'. Hence more needs to be included about the shaving process and how it was applied in this experiment. I got the impression that from Gallet et al. (2009) the shaving/smearing of the surface grains by the spatula (especially when close to freezing) could have had an impact on surface optical reflectance. I expect this not to be the case in cold labs at -15 degrees Celsius, but it requires a more detailed discussion about how preparation of IC sample surfaces effect SSA. Discussions at the Dayos Grain Size Measurement workshop in 2014 and my own experience of making IC measurements suggest that the SSA from IC is (thankfully) not very sensitive to sample preparation. The pressure required to cause sintering as part of the sampling process is highly unlikely to be achieved. Rather, making sure the sample container is completely full by addition of snow to fill any gaps in the extracted sample, and light compaction of snow to be flush with the container surface is preferable so that reflectance is less likely to come from the edges of the sample container. This negligible impact of sample preparation appears to be shown in the comparison of distributions of IC + particles and IC – particles in Figure 2, where distributions overlap. As both field experience and results in Figure 2 contradict the message that sample surface preparation is crucial, this message needs to be revisited.

We included a detailed illustration of our sampling procedure to clarify the individual sampling steps (Fig.4). Additionally, we expanded the manuscript text accordingly. The criticised steps have always been taken into account.

Unfortunately, we can not agree with this comment. As we show with the "nature-identical new snow" (we fixed the terminology issue as it is more a sprayed, artificial snow), which has the largest specific surface area of the tested samples, there is no difference between the SSA measured with the IceCube and the micro-CT in case of loosely sintered snow (and this is also shown in the Greenland sample, where the loose topmost layer has identical SSA for both methods). We elaborate on the sampling procedure of the surface in more detail. All surfaces of the samples were tilted before the first IC measurement (when cutting off the protruding snow), and for the loose particle collection, tilted again and gently taped over a Petri dish. As shown in our article, we could not find a parameter that predicts large differences, except that fragile snow seems more prone to produce very small particles.

Manuscript Fig.4. Step-by-step illustration of the sampling procedure. (a) within each snow block (snow types A, B, C and D), we identified a homogeneous layer of snow with SMP measurements. We removed the unsuitable material above the sampling layer with a saw. Afterwards, we gently brushed the surface to remove of loose particles from the sawing process. (b) IC sample extraction with the IC sampling device (piston). The snow sample is 35 mm thick. (c) transfer of the snow sample into the IC sampling holder, which has a height of 30 mm. (d) transferred snow sample with 5 mm of protruding snow. (e) we cut off the protruding snow with a sharp spatula (the sample is slightly tilted) following (Gallet et al., 2009) and perform the first IC measurement (IC + particles). (f) we tilt the IC sampling holder above a Petri dish (blue) to collect the remaining loose particles created during the sampling step (e) for further analysis (macro-photographs and particle weight) and the sample is measured again with the IC (IC - particles).

Serious consideration needs to be made as to whether relative percentage difference is a fair way to present the results, particularly when the mean or median values range from <10 to  $>50 \text{ m}^2 \text{ kg}^{-1}$ . I suggest presenting the measurement uncertainty in m2 kg-1 is more appropriate, e.g. a bias or RMSE. This is illustrated by Figure 2, where the actual difference between extents of upper and lower quartiles between CT out of IC / CT reference and IC respectively, either overlap for type C and E, or are approximately 2-3 m2 kg-1 apart for type A and D. And when the four distributions of CT are considered against the two IC distributions, overlap of distributions is more common than not. Some discussion about what level of natural SSA variability might be expected within a sample (CT or IC) needs to be added here. Depending on the orientation of the sample in IC measurement I would expect variability in spectral reflectance, particularly in snow types that are not highly homogenous in structure, size and orientation. Hence SSA variability of the order 2-3 m2 kg-1 may well be within measurement noise. As an exemplar, Fig 2 shows that differences between distributions of CT surface and CT mid are on the same order of similarity to the difference between IC + particles and CT out of IC. While I expect CT mid to be the best measurement to compare other measurements to, the fact that there is such spatial variability within a CT sample, suggests that the comparison between CT and IC is not drastically worse than the within CT measurements. Can this be discussed in further detail as it appears to add sensible uncertainty caveats to one of the headline conclusions, which is there is a SSA difference of 20-52% in the 5-25 m2 kg-1 range when measured by IC and CT.

We included the MBE in addition to the relative percentage difference. We did not write in the paper that all snow types are biased, but we found this behaviour for certain snow types, and especially in the Greenland snow.

The visual and statistical comparison of distributions (Fig 1 and 2) is good, but this raises a concern at the low number of sample values (Table 1 shows n=1-8) which make up these distributions. I appreciate the time required to make CT measurements, so this not being a high n-value is understandable. However, how was the n value calculated for IC? Was it a single sample measured in different orientations? Or were there a number of samples in the same snow layer? Considering that the IC is designed for field use and implications of results increasingly tend to be considered in recent literature when using larger distributions (n >10) of measurements in similar layers, these are very low sample numbers to be making robust conclusions. However, there is a balance to be struck here. These initial results are useful for the community to see, but I think that it points the way for future work, rather than being definitive about the applicability of IC measurements at SSA <30 m2 kg-1. It might be a suggestion for a revised version of this manuscript to be a brief communication rather than a full article in TC?

We think that the reviewer misunderstood our sampling. The samples consist of homogenous snow, as shown with the SMP measurements. Our statistical tests show a good level of significance, and the trend of overestimation for most snow types for all aggregated measurements is clear and significant. We want to keep our study as a full article as a brief communication does not provide the scope to elaborate on all the details, especially with the adaptations we made in response to the reviews.

The application of TARTES to provide an explanation of potential impact of the surface grain size on optical reflectance is good. However, as a non-expert, does the choice of a 1m thick snow substrate matter, rather than a sample thickness of 300 mm which is a common depth of the IC sample container?

The sample thickness is 30 mm. As the wavelengths of interest only penetrate the surface area of the snow sample (i.e., the first few millimetres), a substrate thickness of 1m is adequate for our simulations.

What is nature-identical new snow (largely  $>30 \text{ m}^2 \text{ kg}^{-1}$ ) in the context it is presented? Figure 1 looks like the distribution of IC + particles (more realistically how samples would be measured in the field) are very similar to the CT distribution, so we need more detail on what is 'nature-identical'. It suggests the rest of the samples are dissimilar to nature, which is a worry when drawing implications from this experiment. This may just be a terminology issue, but it needs to be addressed.

We deleted the term "nature-identical snow" - as this was sprayed snow (so more like technical snow, consisting of small spheres). We changed the terminology to artificial snow and elaborated on the production details in the new version of our manuscript.

Rather than a list of minor comments, at this stage I would encourage the authors to:

- Expand on the introduction to contrast IC and CT to a broader range of microstructural measurements and implications for their use. We expanded the introduction and included a data set from Greenland to show the issue we are investigating.
- 2. Increase the clarity of the 'hypothesis statement' at the end of the introduction (i.e. this paper does, this, and this, and this...) Done.

- 3. Add clarity on what is being presented in the box plots (are they median and IQR, or mean and standard deviation as in Table 1). The data set is available online (doi:10.16904/envidat.333.), and we adapted the figure caption accordingly.
- 4. Add more details throughout (as per issues raised above), and in the discussion section add more on the implications of these results for a) field measurement using IC, and b) what using measurements from IC may mean for applications where SSA is crucial. We designed our study to investigate the artificial particles as a source of an SSA overestimation with the IC. We want to raise awareness in the snow science community for this issue rather than present a final solution. Further research is necessary to study possible solutions for this matter which would go beyond the scope of this paper.
- 5. Check the cross references are correct (e.g. 'Tab 3' is cross-referenced, but is not in the main manuscript). Done.

We tried to answer all the points in the adapted version of our manuscript.

**Response Review RC2**

The paper is concise and relatively clear though the format is a bit disorganized and the terminology is not always consistent. The figures positively contributed to the representation of the data and methodology though more specific figures related to sample preparation would improve the explanation of this process.

We addressed this by adding a detailed step-by-step illustration of the sampling process as well as detailed information in the manuscript (manuscript Fig. 4).

The experimental design was thoughtful and could be impactful to the field. Refining our understanding of the caveats associated with field instruments will help improve future measurements and use. The paper would benefit if the authors add more information about sample preparation (with additional visual aids) and context for the snow samples used in the experiment. It would also be beneficial to describe the micro-CT methods and analysis, including any differences associated with two nominal scan resolutions (15 &18um). Could the authors also better describe the "manufactured micro-CT sampling kit" and how it is used?

We improved the micro-CT sampling kit figure and provided a more detailed description.

Manuscript Fig.5. Illustration of the micro-CT sampling kit with sample holder in yellow and piston in black. The snow sample is coloured in white. (a) is the manufactured micro-CT sampling kit with the piston imitation (35 mm depth) to retrieve the snow sample. (b)

illustrates the snow sample transfer into the sampling holder (30 mm depth), and (c) shows the cutting process of protruding snow to create a flat sample surface.

Generally speaking, it would be helpful to have the results presented in the same order as the data, or it would help to introduce and explain the format in which the data will be presented. Figures would be more accessible if they were adjacent to where they are references in the text and the figure captions could be improved if more information is added to them and if they are written in complete sentences.

Line 6: Please describe what these "artificially created particles" are. We adapted the sentence to improve clarity. *We suspected that particles artificially created during the sample preparation were the source of the difference.*

Line 20: I recommend re-writing the sentence "We focus on.." to improve clarity. We adapted the sentence to improve clarity. We focus here on artificial particles occurring on the snow sample surface produced with the mechanical IC sampling procedure as a source of the discrepancy.

Line 22: Is it necessary to mention black carbon at all? It is the only time it is mentioned in the entire paper. Agreed.

Line 39: Recommended change "Next, a second SSA is measured with ..." Done. Next, we perform a second IC SSA measurement (same sample) without loose particles on the sample surface (Fig.6 IC - particles).

Line 46-48: Can you comment on any variability associated with different micro-CT scan resolutions?

There is no variability associated with the different resolutions in our micro-CT scans. The minimum resolution needs to be 3 times the object size (for 15µm this is 45 µm, for 18µm this is 54µm, respectively). The smallest scanned particles are at least 77 ± 49 µm (results grain survey Tab.1).

Line 54: I recommend rewording the sentence "Eight pictures..." for improved clarity. Done. To obtain the grain size distribution, we randomly choose eight pictures with about 1000 grains in total for snow types B, D, and E.

Line 68: I recommend rewording the sentence "As the desired output variable.." for improved clarity.

Done. To calculate the SSA, we needed to apply a conversion to the diffuse albedo data set. Hence, we produced a synthetic SSA data set for both wavelengths, which is used for a poly-fitting procedure. We deem a five-degree polynomial adequate to describe the relationship between the calculated diffuse albedo and SSA.

Line 71: In the results section, Table 3 is both spelled out and abbreviated as "Tab.3". Fixed.

Line 79: Recommend changing sentence to " However, the SSA measured by the micro-CT is 24% smaller than that measured by the IC." Done.

Figure 1 caption: The figure is described as having a "left, middle, and right". Do you mean top, middle, and bottom? It might be better to refer to "panels (a), (b), and (c)" instead. Fixed.

Figure 3: Please describe the manufactured micro-CT sampling kit. There is no reference to the (b) and (c) labels on this kit.

We adapted the figure and the caption to better illustrate the micro-CT sampling kit. We fixed the label references.

**List of relevant changes**

General

- reorganization manuscript to improve the flow of reading
- active voice
- cross-references were checked and fixed
- improved terminology for clarification
- figure captions expanded
- introduction expanded
- discussion expanded
- uploaded Greenland data set and artificial snow data set (doi:10.16904/envidat.333.)

**Introduction**

- expanded introduction
  - described and included a new data set (Greenland summit 2012, Schneebeli et al., 2015) as motivation for our study
  - o added new figure: Figure 1

**Methodology**

- information about artificial snow production and reference (Brandt et al., 2011)
- added grain size photographs for all snow types (new figure: Figure 2)
- added new data to improve the description of our snow types
  SMP data set and figure (new figure: Figure 3)
- step-by-step illustration of the IC sampling procedure (new figure: Figure 4)
- micro-CT sampling kit illustration (new figure: Figure 5)
- added micro-CT to improve the description of our snow types (Tab.1)
- detailed description of IC and micro-CT sampling kit procedure

**Results**

- added MBE calculation
- added micro-CT densities (Tab.1)

Discussion

- expanded discussion for wavelength (response to CC1 comment)

**Conclusion**

- added outlook

---

## Author Response (AR2)

**Author Response**

Paper # https://doi.org/10.5194/egusphere-2022-501

**Impact of the sampling procedure on the specific surface area of snow measurements with the IceCube**
*Martin and Schneebeli*

**Editor comments**

1. Section 2.1. Please specify where these samples were taken from. If all alpine please state this. The new figure 2 and associated density information allow the readers to identify samples where this might be a problem, and figure 1 demonstrates this is indeed happening outside alpine regions.
*Done. We included the word "alpine".*

2. Line 52: Chapter -> section, also figure 2 caption
*Done. Both were changed to "section".*

3. Figure 2: are these the discarded fragments as indicated in line 52? If so, please state in the caption and discuss that these may not be representative of the crystals being observed.
*The pictures show the original snow grain and the artificial particles. We adapted the figure caption accordingly.*

4. Figure 6. Please remove 'nature identical snow' from the legend. Please also indicate in the caption whether the box plots represent median and interquartile range or mean and standard deviation, as per reviewer 1 comment.
*Done. We changed "nature identical snow" to "artificial snow". We added "median and interquartile" in the figure caption. Additionally, we included more information in the caption.*

5. Line 37. Please provide a few more details on the production of artificial snow rather than just including the reference. For example, it would be useful for the reader to know that 'this was sprayed snow (so more like technical snow, consisting of small spheres)'. Please clarify 'We did not use the artificial snow type in our experiment to directly compare the IC and micro-CT SSA measurements.' The comparison in Fig 6a is between two completely separate samples of artificial snow prepared in the same way?

*Done. We included more information about the production process and clarified the sentence. The artificial snow was not used for our multi-step sampling strategy but was only measured with the IC and micro-CT. We included a technical report as a detailed reference.*

6. Line 113. Consider an alternative to 'point out' e.g. protrude, jut out, stick out
*Done. Changed to "protrude".*

7. Line 142. Please define 'MBE' acronym, and potentially include an equation if non-standard term.
*Done. We defined Mean Bias Error (MBE)*

*Additionally, we corrected the Journal abbreviations in the bibliography and corrected minor wording issues.*